# Is un stylo sharper than une épée? Investigating the interaction of sound symbolism and grammatical gender in English and French speakers

**David M. Sidhu**[1]*, **Penny M. Pexman**[1], **Jean Saint-Aubin**[2]

**1** University of Calgary, Calgary, Alberta, Canada, **2** Université de Moncton, Moncton, New Brunswick, Canada

* dmsidhu@gmail.com

**Data Availability Statement:** All data files are available from the OSF database (DOI 10.17605/OSF.IO/YQVD9).

## Abstract

While the arbitrariness of language has long been considered one of its defining features, there is growing evidence that non-arbitrariness also plays an important role. Here we investigated two sources of non-arbitrariness: systematicity (via grammatical gender) and iconicity (via shape sound symbolism). We manipulated these two elements orthogonally, allowing us to examine the effect of each. In Experiment 1, we found that French speakers associated nonwords containing feminine (masculine) endings with round (sharp) shapes. French speakers also associated nonwords containing round-sounding (sharp-sounding) phonemes with round (sharp) shapes. This was repeated using auditory presentation with both an English-speaking (Experiment 2a) and French-speaking (Experiment 2b) sample. As predicted, the English speakers showed no effects of grammatical gender, while the French speakers did. These results demonstrate that speakers of a language with grammatical gender associate different properties to words belonging to different genders. The results also show that sound symbolism can emerge in stimuli with existing associated information (i.e., endings indicative of grammatical gender, and the association that they evoke). Finally, while previous studies have looked at effects of arbitrary and non-arbitrary mappings contained in a single stimulus, this is the first study to demonstrate that different kinds of non-arbitrary mappings can have an effect when appearing in the same stimulus. Together these results add to our understanding of the importance of non-arbitrariness in language.

## Introduction

### Arbitrariness, iconicity and systematicity

Does the form of a word provide cues to its meaning? Moreover, are individuals sensitive to these cues? These are fundamental questions about the nature of language that cut across both linguistics and cognitive psychology. One possibility is that form and meaning are arbitrarily related, such that form provides no reliable cue to meaning. Indeed, this has often been taken

**Funding:** Funded by Natural Sciences and Engineering Research Council (NSERC) of Canada Discovery Grant RGPIN 217309-2013 to PMP; RGPIN-2015-04416 to JS; Canada Graduate Scholarship CGSD3 476111 2015 to DMS. Alberta Innovates Health Solutions Graduate Scholarship 201500125-1 CA# 3874 to DMS.

**Competing interests:** The authors have declared that no competing interests exist.

as the default. For instance, Charles Hockett suggested that the arbitrary relationship between form and meaning (i.e., one lacking any "physical or geometric resemblance", ([1], p. 8) was one of the universal features of language. Dingemanse, Blasi, Lupyan, Christiansen and Monaghan defined arbitrariness as cases where "apart from a social convention to use word A for meaning B, there is no connection between the sound of a word and its meaning" ([2], p. 604).

More recently, arbitrariness has come to be understood as *one possible* type of form-meaning mapping (for a review see [2]). Indeed, there seem to be other types of form-meaning mappings that are non-arbitrary. In this paper we use the framework suggested by Dingemanse et al. [2], in which there are two (non-exclusive) types of non-arbitrary mappings: iconicity and systematicity. In iconicity, aspects of form map onto aspects of meaning via resemblance. Take for instance the word *ding*, whose abrupt onset and fading offset resemble aspects of its meaning (i.e., these same properties in the sound of a bell; [3]). This would be considered an instance of *onomatopoeic* iconicity: instances in which the sound of a word directly resembles some auditory aspect of its meaning.

It is also possible to have *non-onomatopoeic* iconicity, in which the sound of a word resembles non-auditory aspects of its meaning indirectly. One way in which this can happen is through the *sound symbolic associations* of phonemes: various perceptual or semantic features that individuals associate with particular phonemes (for reviews see [4, 5]). One example is the tendency to associate certain phonemes with visual roundness, and others with visual sharpness (i.e., *the maluma/takete effect*; [6]). In particular, studies indicate that individuals associate sonorant consonants (e.g., /l/, /m/ and /n/), voiced stop consonants (e.g., /b/, /d/, /g/; though to a lesser extent), and back (often rounded) vowels (e.g., /u/), with roundness. Conversely, individuals associate voiceless stop consonants (e.g., /p/, /t/ and /k/), and unrounded front vowels (e.g., /i/) with sharpness [7, 8]. To illustrate how these associations might contribute to non-onomatopoeic iconicity, consider the word *balloon*. It refers to a round object and contains phonemes that are sound symbolically associated with roundness.

Individuals appear to be sensitive to iconic form-meaning mappings. There is evidence that iconic words are easier for both infants [9] and adults [10] to learn. In addition, neuroimaging studies with adults have suggested that iconic words are processed differently than non-iconic words, leading to observable differences using both fMRI (e.g., [11]) and EEG (e.g., [10, 12, 13]). There is also behavioural evidence of a processing benefit for iconic words [14, 15]. Lastly, many studies with nonwords have shown that they are responded to differently when presented with iconically congruent vs. incongruent stimuli (e.g., [16–18]).

Despite this work, the extent to which non-onomatopoeic iconicity can affect processing is still somewhat unclear. Studies that have directly examined sound symbolic associations such as the maluma/takete effect in existing language have been equivocal [5, 18–20] with some authors speculating that when linguistic stimuli are associated with existing semantic information, this might diminish effects of sound symbolism (see [18, 21]). That is, when a word has meaning associated with it, the activation of that meaning during processing could interfere with sound symbolic associations evoked by its phonemes. One goal of the present study was to extend this previous work by examining the extent to which the maluma/takete effect would emerge in nonwords with associated information from word endings that are indicative of grammatical gender. In a departure from previous studies, this associated information is implicit and not situated in word meanings but rather associated with the grammatical gender categories to which the words ostensibly belong.

The other possible type of non-arbitrary form-meaning mapping is *systematicity* [2]. This refers to statistical regularities among groups of words belonging to the same syntactic (or even semantic) category. For instance, there are systematic differences in the forms of nouns and verbs [22–24]. Farmer et al. [22] examined the phonological forms of 3,158 nouns and

verbs, and found that nouns were more similar to each other than they were to verbs (and vice versa for verbs). Moreover, individuals were sensitive to these cues, responding faster to a phonologically typical noun or verb when the sentential context led to the expectation of a noun or a verb, respectively. Systematicity could even apply to broad semantic categories. For instance, studies have demonstrated that there are differences in the forms of concrete and abstract nouns, and that participants are more accurate when judging the concreteness of phonologically typical vs. atypical, concrete and abstract nouns [26]. Beyond these examples, Dingemanse et al. noted that derivational morphology represents "another pervasive source of systematicity" ([2], p. 607). For instance, the word *teacher* reveals an association with the word *teach*.

Importantly, words do not fall wholly into the categories of arbitrary, iconic or systematic. Rather, words can contain each kind of mapping to varying degrees. In fact, studies have begun to demonstrate that individuals are sensitive to different kinds of cues appearing in the same word. For instance, Monaghan, Christiansen and Fitneva [26] found that there are advantages to having both arbitrary and systematic elements within individual words. Participants were better able to learn nonwords' meanings when they contained both arbitrary and systematic elements, as compared to those that were wholly arbitrary or systematic. However, whether speakers can be sensitive to both systematic and iconic cues (when separate; cf. [27]), within the same stimulus, remains to be seen. The present study examined this in the context of grammatical gender.

## Grammatical gender

In many languages, nouns belong to particular classes commonly referred to as grammatical genders [28]. These affect the forms of other elements of the sentence (e.g., the forms of articles or verbs might then have to agree with the gender of the noun; [29]). Despite their name, grammatical genders do not necessarily delineate based on gender per se (e.g., the Georgian language makes a distinction based on animacy). Nevertheless, many languages–such as French–do classify nouns into categories designated as feminine or masculine. In French, nouns that have a biological sex are typically assigned to a category based on that sex (e.g., a male singer is denoted by the masculine noun *un chanteur*, while a female singer is denoted by the feminine noun *une chanteuse*). However, delineations are not purely based on gender, as nouns that do not have a biological sex are still assigned to one of these two categories. This assignment does not necessarily reflect semantic features; for instance, *chair* is feminine (*une chaise*) while the semantically similar *stool* is masculine (*un tabouret*).

Speakers of a language with grammatical gender are constantly required to attend to the gender of nouns, in order to use articles and pronouns that agree with those nouns grammatically. This leads to the possibility that a given noun's representation may be influenced by its grammatical gender (see [30]). Vigliocco, Vinson, Paganelli and Dworzynski [31] proposed that because words for females and males belong to feminine and masculine grammatical gender categories, individuals may come to associate either grammatical gender category with properties typically associated with females and males. Once this happens, nouns belonging to either grammatical gender category–even nouns without a biological sex themselves–may become associated with typically female and male traits (for other perspectives on the development of such effects, see [32–34]).

Indeed, there is evidence of individuals extending gender stereotypical features to words based on their grammatical gender. Boroditsky et al. [32] compiled a list of objects that had contrasting genders in German and Spanish. They then asked German and Spanish speakers to describe each object with the first three adjectives that came to mind. Independent raters

judged these adjectives as being feminine or masculine. By including objects that belonged to different genders in German and Spanish, the authors were able to demonstrate that the same object was described by stereotypically female- or male-associated adjectives, depending on its gender in a language. For instance, *bridge* is feminine in German, and was described by German speakers as "beautiful", "elegant" and "fragile". It is, however, masculine in Spanish, and was described by Spanish speakers as "big", "dangerous" and "strong" (see also [35]). Furthermore, studies using a semantic differential technique (i.e., ratings on scales anchored by antonyms such as pleasant/unpleasant) have found that a given noun is rated higher in potency (i.e., *strong vs. weak)* when it is grammatically masculine vs. feminine [36–38].

Grammatical gender categories often show signs of systematicity. That is, there are statistical regularities among groups of words in each gender category. For instance, Lyster [39] examined a corpus of 9,961 French nouns, and discovered that 81% of feminine nouns, and 80% of masculine nouns, have orthographic endings that are predictive of their grammatical gender. This is a prevalent example of systematicity in language. Studies have capitalized on this to examine whether these cues in isolation (i.e., predictive endings attached to nonword stems) can lead to the sort of effects observed by Boroditsky et al. [32]. This approach has the advantage of controlling for possible cultural effects in the use of real words. For instance, Boroditsky et al. [32] supposed that "the way objects are personified in fairy tales or poetry may depend on the grammatical genders of their names"; this could affect their representation indirectly.

Ervin [40] found that Italian speakers rated nonwords with a feminine ending (-*a*) as being prettier, weaker, and smaller than those with a masculine ending (-*o*). In a more recent study, Vuksanović, Bjekić and Radivojević presented Serbian-speaking participants with nonwords that were "indicative [of] two different grammatical genders" ([41], p. 387) in Serbian. These were ostensibly the names of instruments, and participants were asked to rate those instruments on a variety of dimensions. As expected, instruments labeled with feminine (masculine) nonwords were judged to be higher on scales typically associated with females (males). This was even observed when participants were shown an image of each instrument.

However, examining the nonwords used by Vuksanović et al. [41] reveals a potential confound. Besides differences in endings, the feminine and masculine nonwords differed systematically in their stems. Every feminine nonword stem contained a sonorant consonant, while none of them contained a voiceless stop. Conversely, only two of the eleven masculine stems contained a sonorant, while nine of them contained a voiceless stop. Given the different sound symbolic associations of these different groups of phonemes (e.g., [42, 43]), it is possible that these drove the gendered associations rather than the endings themselves. In addition, sound symbolism may have played a role in the study by Ervin [40]. While nonwords ending with -o were rated as larger than those ending with -a, recall that /o/ is sound symbolically associated with large objects [44, 45]. This presents the intriguing possibility that sound symbolism may be an as yet unmeasured contributor to grammatical gender effects.

Indeed, grammatical gender effects have proven to be susceptible to a variety of moderating factors including: age (in some studies the effect only emerges in older infants; e.g., [35, 46]); language (some studies suggest the effect to be stronger in languages with two vs. more than two grammatical genders, perhaps because the former makes gender association clearer; see [31]), animacy (in some studies the effect emerges only for animate nouns; e.g., [31]), gender (stronger effects for feminine targets; e.g., [47]), and whether the stimuli are words or pictures (in some studies the effect only emerges for words; e.g., [48]). Thus, another purpose of the present study was to test for grammatical gender effects while also accounting for potential confounds of phonology.

## The present study

As noted earlier, certain phonemes are sound symbolically associated with either round or sharp shapes. Furthermore, in a study examining whether the maluma/takete effect would extend to existing first name labels, Sidhu and Pexman [49] found that female (male) names were more likely to be associated with round (sharp) shapes. This has since been replicated with a French-speaking population [50]. The results of these studies suggest that gender is also associated with round and sharp shapes. Thus, shape provides an excellent opportunity to test the coexistence of iconic cues (i.e., shape-sound symbolism) and systematic cues (i.e., endings indicative of grammatical gender), within individual linguistic stimuli. As mentioned earlier, while arbitrariness, iconicity and systematicity are believed to be able to coexist, even at the level of single words, this has yet to be fully examined for iconicity and systematicity. In the present study, we examined whether individuals are able to attend to, and be affected by, both kinds of cues in a single stimulus.

## Experiment 1

### Method

**Ethics statement.** Experiments 1 and 2b were approved by the Université de Moncton research ethics board; Experiment 2a was approved by the University of Calgary research ethics board. An informed consent form was used in all experiments.

**Participants.** Participants were 50 undergraduate students (40 female; *M* Age = 19.26, *SD* = 1.60) at the Université de Moncton who received course credit. Note that the age of one participant was not collected. All participants reported French fluency and normal or corrected to normal vision.

**Materials and procedure.** The linguistic stimuli were 24 nonwords created in the following manner. Half of the nonwords contained a round-sounding stem (e.g., *mon-*), and half contained a sharp-sounding stem (e.g., *tip-*). Round-sounding stems consisted of voiced stops (/b/), sonorant consonants (/l/, /m/ and /n/) and back and/or rounded vowels (/ɑ/, /u/, /y/, /œ/, /ɔ/ and /o/); sharp-sounding stems consisted of voiceless consonants (/p/, /t/ and /k/) and front unrounded vowels (/i/ and /e/). These types of phonemes have been shown to be associated with round and sharp shapes, respectively, in previous studies [8, 9]. Half of the nonwords with each type of stem were attached to an ending that typically occurs ($\geq$ 95.00%; [51]) in feminine French nouns, and half to an ending that typically occurs in masculine French nouns ($\geq$ 94.00%; [51]); see Table 1. Thus, there were six each of: round-feminine, round-masculine, sharp-feminine, and sharp-masculine nonwords. Notably, none of these endings contained round- or sharp-sounding consonants. Within each group of nonwords, two to four nonwords had an ending with a round vowel. Since this was not perfectly balanced, the vowel content of

**Table 1. Grammatical gender endings used in Experiment 1.**

| Feminine Endings | Masculine Endings |
|:---:|:---:|
| -arde | -age |
| -esse | -ard |
| -euse | -eau |
| -ier | -eur |
| -oise | -eux |
|  | -ier |
|  | -is |

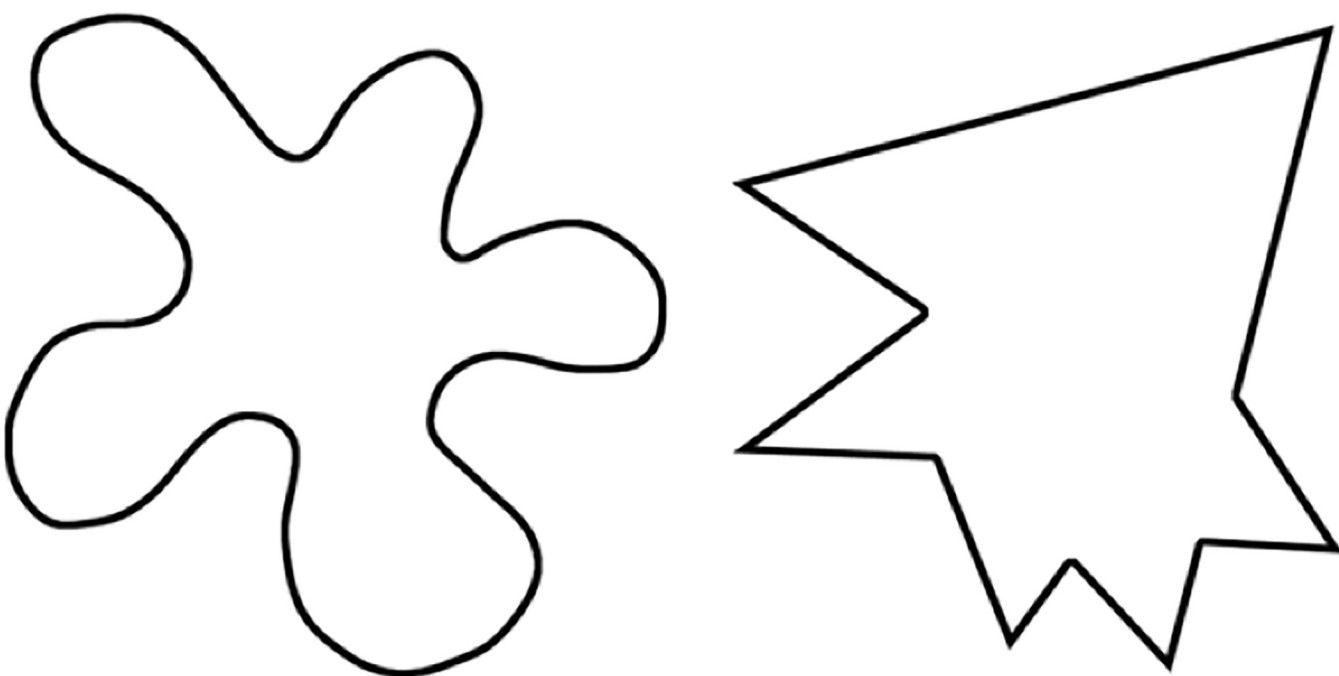

**Fig 1. Examples of round and sharp stimuli used in each experiment.**

endings was included as a control variable in the analyses. Stimuli lists for all experiments are available at the following OSF repository: https://osf.io/yqvd9/.

We first ran a pilot study in order to verify that nonword gender would be apparent based on nonword endings. In this study, 20 additional French-speaking participants (19 female; $M$ age = 18.91; $SD$ = 1.45) assigned a gender to each of the nonwords by selecting *un* or *une* as an appropriate article. All nonwords were responded to accurately (i.e., with an article that matched the intended gender) at least 80% of the time ($M$ = 95.83, $SD$ = 6.63). Accuracy was not significantly different for nonwords with feminine ($M$ = 94.44, $SD$ = 7.55) vs. masculine ($M$ = 97.22, $SD$ = 5.55) endings, $t(22)$ = 1.03, $p$ = .32; nor for those with round-sounding ($M$ = 97.22, $SD$ = 5.55) vs. sharp-sounding stems ($M$ = 94.44, $SD$ = 7.55), $t(22)$ = 1.03, $p$ = .32.

The shape stimuli consisted of 48 shapes created using Adobe Photoshop. Half of these were comprised of round edges and the other half were comprised of sharp edges. The sets of round and sharp shape stimuli were matched in terms of area, height and width. These shapes are typical of those used in studies on the maluma/takete effect (e.g., [19, 52]). See Fig 1 for examples.

Each trial began with a fixation cross for 1000 ms. This was then replaced by a visually presented nonword that remained onscreen for 2000 ms, which the participants were instructed to read to themselves silently. After this, participants were shown a pair of shapes: one round and one sharp, one on the left and one on the right side of the screen. Alignment of the round and sharp shapes for particular nonwords was counterbalanced across participants. Participants chose the shape that they thought best matched the nonword via keyboard press. Their response triggered a 500 ms blank screen, after which the next trial began. There was one practice trial, followed by 24 trials in the experiment proper.

## Results

Analyses for all experiments consisted of mixed effects logistic regression models. We used the packages "lme4" [version 1.1-18-1] [53], "afex" [0.23–0] [54], and "RePsychLing" [0.0.4] [55] to

perform our statistical analysis in R [3.5.1] [56]. We took a confirmatory approach (for a discussion see [57]) and fit models including all fixed effects of interest. We developed each model's random effects structure using the approach suggested by Bates, Kliegl, Vasishth, and Baayen [58]. In brief, we began by fitting the model with all random slope terms for each fixed effect and removed correlations among random effects if this did not converge. We then performed a principal components analysis on the random effects and simplified the structure based on the suggested number of components. This was accomplished by iteratively removing the random slope for the highest order effect with the lowest amount of variance [55]. We also tested the inclusion of correlations among random effects, and the effects themselves, using likelihood ratio tests. The detailed procedure for model selection, along with code used for the entire process, can be found in the OSF repository. We only report the results of the model containing the final random effects structure in the text. Note that models always included random subject and item intercepts to deal with non-independence.

In the present experiment, we used this approach to examine the effects of nonword type (round-sounding [1] vs. sharp-sounding [–1]) and nonword gender (feminine [1] vs. masculine ending [–1]) on shape selection, with the likelihood of selecting the round shape as the dependent variable. We also included an interaction between these predictors. The presence of a rounded vowel in the nonword's ending (present [1] vs. absent [–1]) was also included as a control variable. See Table 2 for the resulting model. Results indicated that, compared to the average across all factors, participants were 2.82 times more likely to select the round (sharp) shape for round-sounding (sharp-sounding) nonwords ($p < .001$). They were also 1.23 times more likely to select the round (sharp) shape for nonwords with a feminine (masculine) ending ($p = .02$). The interaction between these predictors was not significant ($p = .43$). See Fig 2. Note that, for this and the subsequent experiment, we also ran a version of the analyses in which endings containing the phoneme /w/ preceding a vowel were also considered round. This did not change the pattern of results.

## Discussion

We observed the maluma/takete effect, with participants associating round- (sharp-sounding) nonwords with round (sharp) shapes. It is notable that this emerged in nonwords containing informative endings (i.e., indicative of grammatical gender), in contrast to typical studies on the effect in which ending does not carry specific information. This suggests that sound symbolism can have an effect even in the context of nonwords with existing associated information. In addition, participants associated nonwords with a feminine (masculine) ending with

**Table 2. Logistic mixed effects regression model predicting round shape choices in Experiment 1.**

| Fixed Effect | B | SE | OR | Wald's Z | p |
|---|---|---|---|---|---|
| Intercept | 0.18 | 0.10 | 1.20 | 1.88 | .06 |
| Type | 1.04 | 0.14 | 2.82 | 7.21 | $< .001$*** |
| Gender | 0.21 | 0.09 | 1.23 | 2.29 | .02* |
| Round Vowel in Ending | -0.05 | 0.10 | 0.95 | -0.46 | .64 |
| Type x Gender | -0.07 | 0.09 | 0.93 | -0.78 | .44 |
| Random Effect | | | | $s^2$ | |
| Subject Intercept | | | | 0.00 | |
| Subject Type Slope | | | | 0.56 | |
| Item Intercept | | | | 0.09 | |

Notes.
* $p < .05$
** $p < .01$
*** $p < .001$

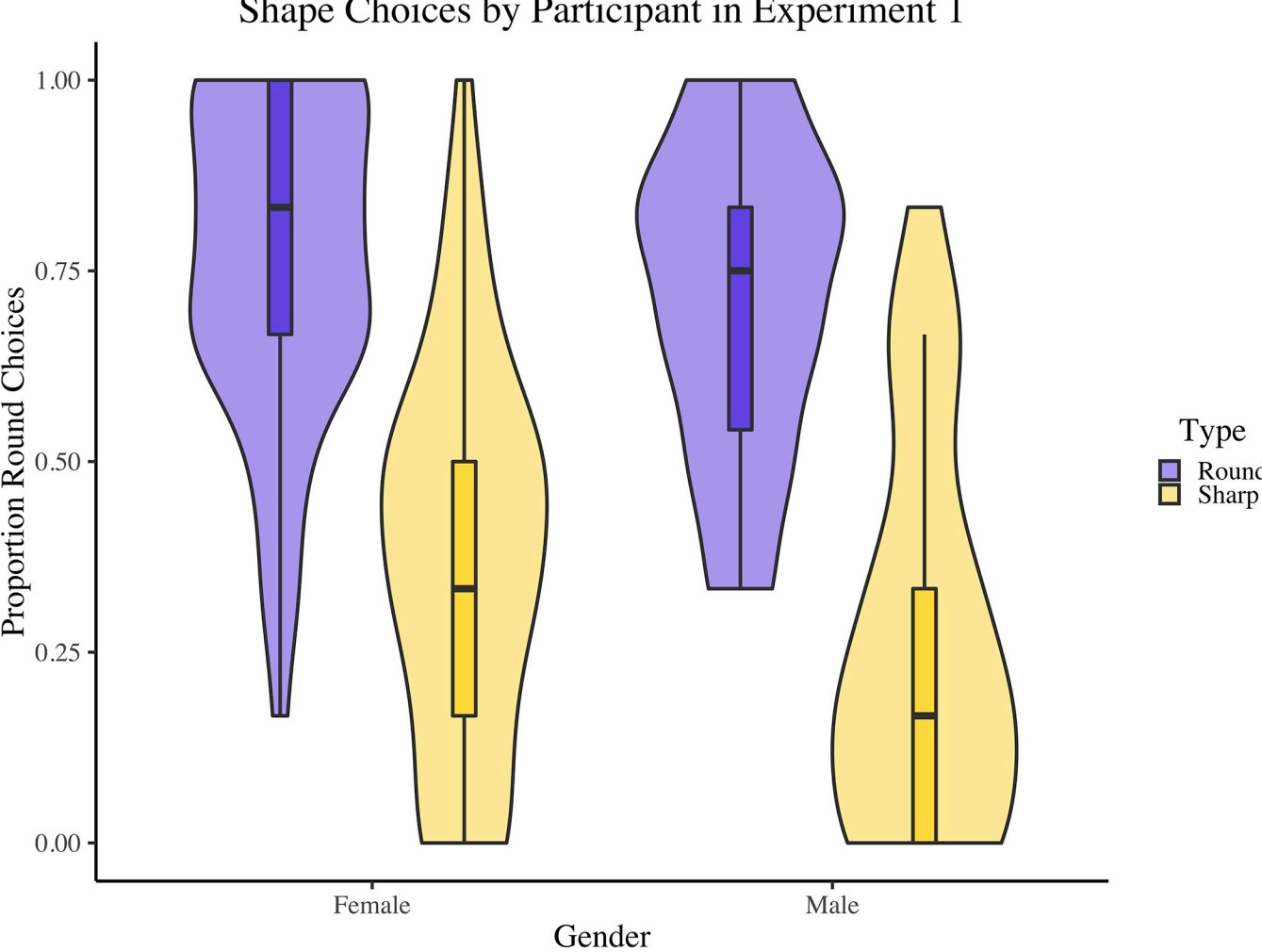

**Fig 2. Proportions of round shapes chosen, by nonword type and gender, for each participant in Experiment 1.** Boxplots represent medians and quartiles; violin plots represent overall distribution of the data.

round (sharp) shapes, suggesting an effect of grammatical gender. Importantly, this result is not explainable as being due to a confound with nonword phonology.

In the next experiment, we tested both English and French speakers, in order to isolate the effects of endings as being due to their grammatical gender per se and not to some uncontrolled factors. That is, if French grammatical gender endings have an effect because of the gender they imply, then we should not see an effect in English monolinguals. In addition, we presented stimuli auditorily, both to ensure that English speakers processed the same phonology as French speakers, and to examine the effects under a different set of conditions (e.g., after removing potential effects of orthography; see [59]).

## Experiment 2a

### Methods

**Participants.** Participants were 74 undergraduate students at the University of Calgary who received course credit. Our stopping rule was as follows: participants were tested until data were collected from 50 participants (44 female; $M$ Age = 20.88, $SD$ = 5.32) who did not

demonstrate French proficiency, defined a priori as accuracy on a French semantic categorization task (does this word refer to a living or a non-living thing?) that did not exceed what would be expected by chance ($p < .01$; [58]). All participants reported English fluency and normal or corrected to normal vision.

**Materials and procedure.** The linguistic stimuli were identical to those used in Experiment 1, except that three nonwords with the ending *-esse* were replaced by nonwords with a different feminine ending. This was done because the suffix *-ess* is used in English to create female forms of words (e.g., *actress*, *goddess*), and we wished to avoid endings with any English gender associations. Once again, these endings did not contain round- or sharp-sounding consonants, and their vowel content was controlled for in the analyses. In addition, nonwords were presented auditorily. The audio files were created using the voice synthesizer of the Acapela group https://www.acapela-group.com/fr/. For the French pronunciation, we used the voice of Claire from France, and for the English pronunciation, we used the voice of Karen from the United States. Audio recordings were created with Sound Tap Streaming Audio Recorder version 2.31, and sound files were edited with WavePad Sound Editor version 5.96. We created two versions of each nonword: one with a French and one with an English pronunciation. Importantly, there were differences between the French and English pronunciations of vowels. For instance, the nonword *boularde* was pronounced /bəlɑrd/ in English, and /bulaʁdø/ in French. Note that after these changes, round-associated vowel phonemes still only occurred in the stems of round-sounding nonwords.

Each trial began with a blank screen for 500 ms. After this, participants were shown a pair of shapes: one round and one sharp, one on the left and one on the right side of the screen. Alignment of the round and sharp shapes for particular nonwords was counterbalanced across participants. Participants were simultaneously presented with the nonword auditorily via headphones. They chose the shape that they thought best matched the nonword via button press on a response box. There was one practice trial, followed by 48 trials in the experiment proper. Participants heard each of the 24 nonwords twice: once with a French pronunciation and once with an English pronunciation. These stimuli in each block were presented in a random order.

Participants then took part in a semantic categorization task with French real word targets, in order to assess their French proficiency [60]. Stimuli consisted of 100 French nouns, half of which referred to living things (e.g., *agneau*: *lamb*) and half of which referred to non-living things (e.g., *acier*: *steel*). Each trial began with a blank screen for 1000 ms. Following this, participants were visually presented with a French word to categorize as living or non-living via button press on a response box. Words were presented in a random order.

## Results

The data from 50 participants, who did not demonstrate French proficiency (where proficiency was defined as above chance performance on the French living/non-living categorization task), were analyzed using the analysis approach described previously. In addition, this analysis also included nonword pronunciation (French [1] vs. English [–1]), as well as additional two-way interactions and a three-way interaction. See Table 3 for the resulting model. Results indicated that, compared to the average across all factors, participants were 2.05 times more likely to select the round (sharp) shape for round-sounding (sharp-sounding) nonwords ($p < .001$). They were also 1.53 times more likely to select the round (sharp) shape for nonwords pronounced with a French (English) accent ($p < .001$). Notably, nonword gender did not reach significance ($p = .26$), nor did any of the interactions. See Fig 3.

**Table 3. Logistic mixed effects regression model predicting round shape choices in Experiment 2a.**

| Fixed Effect | B | SE | OR | Wald's Z | p |
|---|---|---|---|---|---|
| Intercept | 0.06 | 0.09 | 1.06 | 0.62 | .54 |
| Type | 0.72 | 0.10 | 2.05 | 7.52 | < .001*** |
| Gender | -0.08 | 0.07 | 0.93 | -1.12 | .26 |
| Accent | 0.43 | 0.11 | 1.53 | 3.94 | < .001*** |
| Round Vowel in Ending | 0.07 | 0.06 | 1.07 | 1.06 | .29 |
| Type x Gender | -0.02 | 0.07 | 0.98 | -0.33 | .74 |
| Type x Accent | -0.09 | 0.05 | 0.91 | -1.94 | .052 |
| Gender x Accent | 0.02 | 0.05 | 1.02 | 0.50 | .62 |
| Type x Gender x Accent | 0.02 | 0.05 | 1.03 | 0.53 | .60 |
| Random Effect | | | $s^2$ | | |
| Subject Intercept | | | 0.18 | | |
| Subject Type Slope | | | 0.22 | | |
| Subject Accent Slope | | | 0.46 | | |
| Item Intercept | | | 0.05 | | |

Notes.

\* $p < .05$

\*\* $p < .01$

\*\*\* $p < .001$

## Discussion

While monolingual English participants showed a typical maluma/takete effect, they did not show any effect of grammatical gender. This supports the interpretation that the gender results in Experiment 1 were due to the information conveyed by nonword endings.

An interesting outcome of the present experiment was that nonwords pronounced with a French accent, and realized using French phonology, were judged to go with round shapes more often than those with an English accent/phonology. This was unexpected and we can only speculate as to its source. One relevant point is that French contains more rounded vowels than English, and that nonwords realized using French phonology contained a higher percentage of rounded vowels ($M$ = 43.75, $SD$ = 31.40) than those realized using English phonology ($M$ = 32.64, $SD$ = 31.65), $t(23)$ = 2.71, $p$ = .01. However, the effect of accent remains after adding in the percentage of round vowels present in each nonword as a predictor. Note that this was based on an admittedly narrow transcription of the sound files. There was some uncertainty regarding the presence of the vowels /ə/ and /ø/ at the ends of some of our nonwords. Another thing to note is that the same phoneme can be articulated differently by speakers with different accents (see [61]), which could also make them more or less round- or sharp-sounding. For instance, examining the formants of the vowel /u/ in the nonword *bouleuse* revealed that it was pronounced farther back (i.e., with a lower second formant) in the French vs. the English pronunciations. The same was true for other nonwords that shared an /u/ in their French and English pronunciations (i.e., *lumarde* and *mubard*). Recall that vowel backness is associated with roundness.

We next ran the same experiment in a French-speaking sample.

## Experiment 2b

### Method

**Participants.** Participants were 50 undergraduate students at the Université de Moncton who received course credit. Demographic information for these participants was not collected.

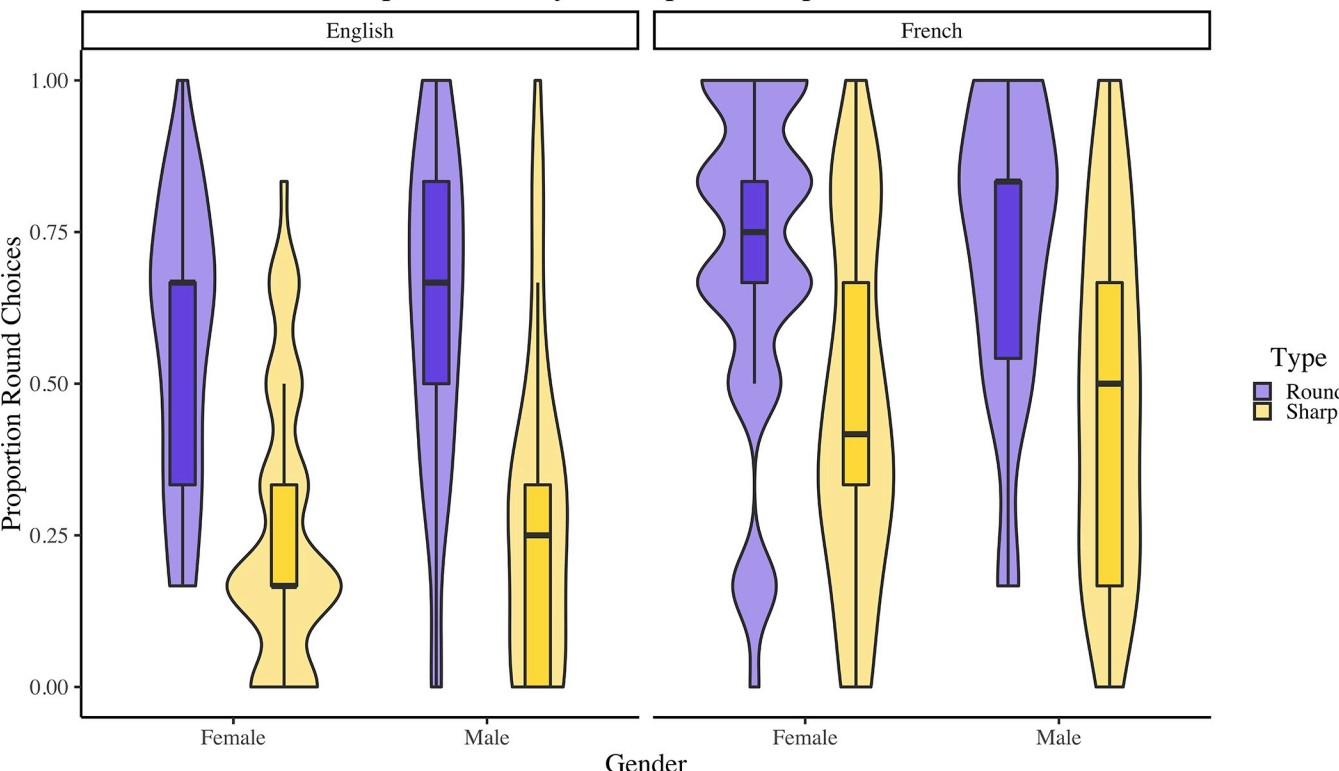

**Fig 3. Proportions of round shapes chosen, by nonword type, gender and accent, for each participant in Experiment 2a.** Boxplots represent medians and quartiles; violin plots represent overall distribution of the data.

All participants reported French fluency and normal or corrected to normal vision. None of them took part in Experiment 1 or the pilot study.

**Materials and procedure.** Materials and procedure were identical to Experiment 2a except that participants made their responses via keyboard press and did not take part in the semantic categorization task.

## Results

The data were analyzed in the same manner as Experiment 2a. See Table 4 for the resulting model. Results indicated that, compared to the average across all factors, participants were 2.54 times more likely to select the round (sharp) shape for round-sounding (sharp-sounding) nonwords ($p < .001$). They were also 1.56 times more likely to select the round (sharp) shape for nonwords pronounced with a French (English) accent ($p < .001$). Additionally, there was a significant interaction between name type and gender ($p = .004$). We followed this up using the "emmeans" [1.2.3] package in R [62] to examine estimated marginal means, and found a significant effect of gender for sharp-sounding nonwords ($p = .007$) but not for round-sounding nonwords ($p = .30$). No other interactions reached significance. See Fig 4.

## Discussion

We once again observed a robust maluma/takete effect, even when using nonwords with informative endings. We also observed the same effect of accent as in Experiment 2a, such that nonwords in a French (English) accent/phonology were more likely to be associated with round

**Table 4. Logistic mixed effects regression model predicting round shape choices in Experiment 2b.**

| Fixed Effect | B | SE | OR | Wald's Z | p |
|---|---|---|---|---|---|
| Intercept | 0.11 | 0.09 | 1.11 | 1.24 | .22 |
| Type | 0.93 | 0.11 | 2.54 | 8.28 | < .001*** |
| Gender | 0.09 | 0.08 | 1.09 | 1.08 | .28 |
| Accent | 0.44 | 0.09 | 1.56 | 5.16 | < .001*** |
| Round Vowel in Ending | 0.02 | 0.07 | 1.02 | 0.33 | .74 |
| Type x Gender | -0.19 | 0.07 | 0.82 | -2.89 | .004** |
| Type x Accent | 0.09 | 0.05 | 1.10 | 1.86 | .06 |
| Gender x Accent | 0.03 | 0.05 | 1.03 | 0.63 | .53 |
| Type x Gender x Accent | -0.03 | 0.05 | 0.97 | -0.54 | .59 |
| Random Effect | | $s^2$ | | | |
| Subject Intercept | | 0.11 | | | |
| Subject Type Slope | | 0.40 | | | |
| Subject Gender Slope | | 0.09 | | | |
| Subject Accent Slope | | 0.25 | | | |
| Item Intercept | | 0.05 | | | |

Notes.

* $p < .05$

** $p < .01$

*** $p < .001$

(sharp) shapes. It is notable that we found this in a French speaking population, suggesting that the effect of accent in Experiment 2a was not a product of the novelty of the French accent to English speakers.

We observed an effect of grammatical gender only for the sharp nonwords, such that nonwords with a feminine (masculine) ending were associated with round (sharp) shapes. One possibility is that the cue to roundness from round-sounding stems overshadowed any effects of grammatical gender. Indeed, there is some evidence of round sound symbolism being more robust than sharp sound symbolism [7, 63, 64]. This might be why the effect of grammatical gender was observed with sharp-sounding stems but not round-sounding stems.

## General discussion

While arbitrariness has historically been considered a fundamental property of language [2], there are important ways in which language can be non-arbitrary. This includes iconicity and systematicity. Importantly, words need not fall wholly into one category or another. Instead, words can contain arbitrary, iconic and systematic properties [3]. Here we tested whether iconic and systematic cues occurring in the same nonword could affect its interpretation. In particular, iconic cues involved phonemes that were sound symbolically associated with either roundness or sharpness. Systematic cues were endings indicative of grammatical gender in French.

In Experiment 1, using a visual presentation, we found that French speakers associated round-sounding (sharp-sounding) nonwords, and those with feminine (masculine) endings, with round (sharp) shapes. In Experiment 2a, using auditory presentation, we found that English speakers (who had little to no knowledge of French) associated round-sounding (sharp-sounding) nonwords with round (sharp) shapes, but as predicted, were unaffected by nonword ending. Interestingly, this experiment also revealed an effect of accent, such that

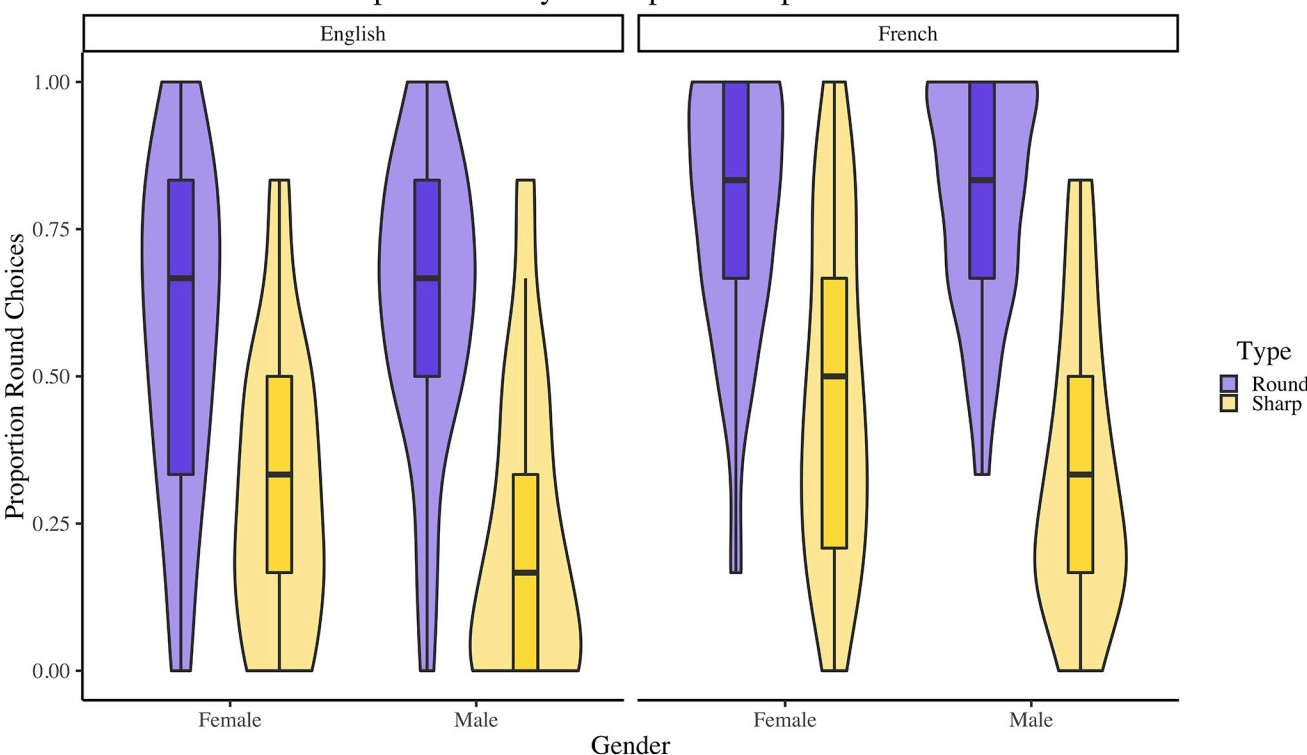

**Fig 4. Proportions of round shapes chosen, by nonword type, gender and accent, for each participant in Experiment 2b.** Boxplots represent medians and quartiles; violin plots represent overall distribution of the data.

participants associated nonwords presented in a French (English) accent with round (sharp) shapes. Finally, in Experiment 2b, we found that French speakers also associated round-sounding (sharp-sounding) nonwords, and those presented with a French (English) accent, with round (sharp) shapes. These participants also showed an effect of grammatical gender for sharp-sounding nonwords, associating such nonwords with feminine (masculine) endings with round (sharp) shapes.

These findings demonstrate that grammatical gender affects interpretation of the meanings of linguistic stimuli. This is in line with the results of a previous study by Vuksanović et al. [41], which also found an association between feminine (masculine) nonwords and roundness (sharpness) when using Serbian endings. However, the present work goes beyond Vuksanović et al. [41] by ensuring that grammatical gender effects were not due to any sound symbolic properties of feminine or masculine endings. We did this by directly manipulating the sound symbolism of nonword stems; and avoiding sound symbolic consonants and controlling for sound symbolic vowels, in the nonword endings. That is, we isolated effects of iconicity (i.e., sound symbolic associations) and systematicity (i.e., grammatical gender). In this context, we still observed effects of grammatical gender, demonstrating that effects of grammatical gender are not explainable via sound symbolism. As suggested by Vilgiocco et al. [31], it seems that traits associated with one gender in the real world can come to be associated with that grammatical gender. Indeed, previous work has shown an association between female (male) names and roundness (sharpness; [49, 50]), although many aspects of these associations (e.g., effects of participant gender) remain unexplored.

This adds to a growing appreciation that the forms of words can impact the meaning individuals will attribute to them (see [2]). Indeed, endings that are predictive of grammatical gender represent a prevalent instance of systematicity, and thus non-arbitrariness in language. Previous work has demonstrated that systematic cues for syntactic class [22] and abstractness [25] affect language processing. This work demonstrates that cues for grammatical gender also affect interpretation.

We also observed a maluma/takete effect across all three experiments. This underscores the ubiquity of the effect, as it emerged in speakers of two languages and in nonwords read with two different accents. Moreover, this finding is notable because we believe it is the first time that the maluma/takete effect has been demonstrated using nonwords with grammatically informative endings. It has been shown that nonwords with inflectional suffixes evoke various associations of the categories they suggest (e.g., [40, 41]). Thus, these results demonstrate that the maluma/takete effect, and sound symbolism more broadly, will emerge even when examined using stimuli with existing associated information. This is notable as some have speculated that when linguistic stimuli are associated with existing semantics, effects of sound symbolism may be attenuated (for discussions see [18, 21]).

While we did observe a maluma/takete effect, if one compares the coefficients observed here to those from a similar study when stimuli did not have existing associated information, there does seem to be anecdotal evidence of attenuation. For example, in Experiment 1b, Sidhu and Pexman [65] visually presented participants with 40 trials of round- or sharp-sounding nonwords to be paired with a target shape. These stimuli did not have grammatically informative endings. When analyzing the first half of trials (i.e., the same number as used in the present experiments) in Sidhu and Pexman [65], the coefficient for the effect of shape was 2.13 ($SE = 0.32$, $p < .001$). When the data from the present Experiment 1 are analyzed in the same manner as in Sidhu and Pexman ([65]; i.e., with shape dummy coded and as the sole fixed effect, with only random intercepts included) the coefficient for the effect of shape is 1.83 ($SE = 0.19$, $p < .001$). Certainly there are other differences between these two studies (i.e., English- vs. French-speaking participants) to consider. Nevertheless, this may suggest that while sound symbolic effects can emerge in the presence of existing information, there may be some attenuation of sound symbolic effects in the face of that existing information. Further evidence of this can be found in Experiment 1a from Sidhu and Pexman [49], in which stimuli were 20 existing round- or sharp-sounding names, to be matched with round or sharp silhouettes. Analyzed in this manner, the coefficient for the effect of shape was 1.18 ($SE = 0.26$, $p < .001$), again suggesting some attenuation. Of course, future research directly comparing stimuli with and without existing information would be needed for any concrete conclusions to be drawn.

One of the goals of the present paper was to examine whether iconic (i.e., sound symbolic) and systematic (i.e., grammatical gender) cues could both have an effect on language users when they appear in a single stimulus. The results of Experiment 1 suggested that this was the case. Previous work had shown that arbitrary and systematic cues within a single stimulus could affect language processing [26]. However, these results are the first demonstration that separate iconic and systematic cues (cf. [27]) can have an effect within a single stimulus. This is an important point, as real words can contain both kinds of cues. For instance, *ball* is a monomorphemic word (i.e., a systematic cue for concreteness; [25]) while containing phonemes that are sound symbolically associated with roundness (i.e., an iconic cue).

Future research should examine the interplay of iconic and systematic cues. There may be some insight to glean from the fact that sound symbolism and grammatical gender interacted in Experiment 2b, with an effect of grammatical gender only appearing in sharp-sounding nonwords. It could be that the association between round-sounding phonemes and round

shapes is stronger than that of sharp-sounding phonemes and sharp shapes. Indeed, there is some evidence of this in the literature (see [7, 63, 64]; cf. [66]). It may be that round shape sound symbolism overwhelmed any effects of grammatical gender, but that sharp shape sound symbolism did not. This suggests that the sources of non-arbitrariness are not always distinct but may interact in the interpretation of a linguistic stimulus. Note, however, that in these studies both cues related to the single dimension on which a decision was being made. This will not always be the case in real language processing. It is also important to point out that iconic cues came before systematic cues in our stimuli, which may have affected how participants weighted the cues.

Notably, iconicity and systematicity are not mutually exclusive. Systematicity refers to large scale statistical patterns in the forms of words belonging to a certain category. In some cases the forms that are indicative of a given category could have an iconic link with that category. For instance, this would be the case if endings predictive of feminine or masculine words contained phonemes that were sound symbolically associated with femininity or masculinity, respectively. However, we assume that even in this case, there are separate forces at work. That is, iconicity still would play a role through form-meaning resemblance, while systematicity might would play a role through regularity in the language. As another example, Imai and Kita [67] theorized that certain sound symbolic associations are language-specific, and are only apparent to individuals when they are present in the iconic words of a speaker's language (e.g., if many nouns referring to round objects contained sonorants). This might be considered an instance of systematic iconicity.

An unexpected finding in Experiments 2a and 2b was that nonwords pronounced with a French (English) accent and with French phonology were associated with round (sharp) shapes. This may have been the cause for the marginal interactions observed between nonword type and accent, in which nonword type had a larger effect in English-accent nonwords. It may be that sharp-sounding nonwords were made somewhat round-sounding with a French accent, thus diluting the effect. As to the cause of the accent effect, one thing to note is that certain orthographic strings were realized with unrounded vowel phonemes in English but rounded vowel phonemes in French (e.g., the first vowel in *bonoise* was /ɑ/ in English but /o/ in French). However, the effect of accent remained after controlling for the percentage of round vowels in each nonword. Nevertheless, this highlights the interesting fact that languages contain different amounts of rounded vowels. For instance, French received pronunciation contains eight rounded vowel monophthongs (i.e., /y, ø, œ, u, o, ɔ, œ̃, and ɔ̃; [68]), compared to three in Standard American Newscaster English (i.e., /u, ʊ, and ɔ/; [69]). It would be informative for future research to explore the effects of differences such as these on the sound symbolic associations of speakers of different languages.

In addition to having different inventories of vowels, the vowels which two languages have in common can be implemented differently in each (e.g., [61]). As mentioned, the vowel /u/ was pronounced farther back (i.e., with a lower second formant) in French vs. English pronunciation. Recent work has demonstrated that sound symbolic effects are moderated by the presence or absence of phonemes in a participant's phonological inventory [70]. The experiments reported here demonstrate that accent could be another important and overlooked moderator of sound symbolism effects. That is, even a simple nonword string like /kiki/ could be implemented differently in different accents, despite containing the same phonemes. It is also important to note that though stimuli with each accent were generated using the same website, they were unavoidably pronounced by different speakers. There may have been differences between the two that contributed to the effect.

These differences also hint at an interesting topic for the field of sound symbolism going forward with regard to cross language comparisons: examining whether these differences are

acoustic or cognitive in nature (or, using linguistic terms, whether they arise from phonetics or phonology, respectively). At present there is some anecdotal evidence supporting both possibilities. Sapir [45] theorized that the reason English speakers did not rate the French phoneme /e/ as small as might be expected (according to size sound symbolism), is that they may have assimilated it to the English dipthong /eɪ/. Additionally, Styles and Gawne [70] theorized that the failure to replicate the maluma/takete effect in speakers of certain languages may have been due to experimental conditions *preventing* assimilation. These suggest the possibility of categorization overriding the acoustic properties of the phoneme itself. On the contrary, Fischer-Jørgensen [71] found that Danish speakers rated several pairs of allophones (i.e., different acoustic realizations of the same phoneme) differently on semantic differential scales. This is an instance of acoustic properties overriding effects of categorization. While future work would be needed to tease out the locus of the accent effect we observed, the possibility that it arises from different realizations of the same phoneme (i.e., different realizations of /u/) supports an effect of acoustic properties. Notably, a supplementary analysis combining Experiments 2a and 2b found that participant language did not interact with the effect of accent ($b$ = -0.01, $p$ = .92), suggesting that participants' phonological inventories did not play a role in the effect.

While the arbitrariness of language had long been considered one of its defining features, there is mounting evidence that non-arbitrariness also plays an important role. Here we showed that both iconicity and systematicity play a role in the processing of linguistic stimuli. Individuals are sensitive to both types of cues, and consider them when making judgments about linguistic stimuli and shape.

## Acknowledgments

The authors would like to thank: Drs. Darin Flynn and Angeliki Athanasopoulou for their assistance transcribing our linguistic stimuli; Marilyne Maltais for her assistance in developing the stimuli and testing participants; Kristen Deschamps and Courtney Miller for their help testing participants; and Summer Abdalla for her help editing an earlier version of this manuscript.

This research was supported by the Natural Sciences and Engineering Research Council of Canada (NSERC) through a postgraduate scholarship to DMS and Discovery Grants to PMP and JS; and by Alberta Innovates: Health Solutions (AIHS) through a graduate scholarship to DMS.

## Author Contributions

**Conceptualization:** David M. Sidhu, Penny M. Pexman, Jean Saint-Aubin.

**Formal analysis:** David M. Sidhu.

**Methodology:** David M. Sidhu, Penny M. Pexman, Jean Saint-Aubin.

**Writing – original draft:** David M. Sidhu, Penny M. Pexman, Jean Saint-Aubin.

**Writing – review & editing:** David M. Sidhu, Penny M. Pexman, Jean Saint-Aubin.

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
