## [Decision Letter · Decision Letter 0]

10 Sep 2019

PONE-D-19-19956

Is un stylo sharper than une épée? Investigating the interaction of sound symbolism and grammatical gender in English and French speakers

PLOS ONE

Dear Mr. Sidhu,

Thank you for submitting your manuscript to PLOS ONE. After careful consideration, we feel that it has merit but does not fully meet PLOS ONE’s publication criteria as it currently stands. Therefore, we invite you to submit a revised version of the manuscript that addresses the points raised during the review process

We would appreciate receiving your revised manuscript by Oct 25 2019 11:59PM. To enhance the reproducibility of your results, we recommend that if applicable you deposit your laboratory protocols in protocols.io, where a protocol can be assigned its own identifier (DOI) such that it can be cited independently in the future. For instructions see: http://journals.plos.org/plosone/s/submission-guidelines#loc-laboratory-protocols

We look forward to receiving your revised manuscript.

Kind regards,

Marcus Perlman, Ph.D

Academic Editor

PLOS ONE

Journal Requirements:

Additional Editor Comments (if provided):

I have sent the paper to two expert reviewers, and I have also read through it carefully myself. The first reviewer generally thought that the study was well done and that the report was clearly written, but also offered some critical comments focused on the framing of the paper. They suggested that major revisions were needed. The second reviewer was highly positive about the paper and made just a few minor suggestions for points where further detail or literature could be added. In my own assessment, I agree with the reviewers that the study was well done, the analysis sound, and the report clearly written. In sum, I thought it was an interesting study that makes a clear and important contribution to the study of iconicity, specifically towards understanding the functions of iconicity and systematicity in words.

The main point of concern of Reviewer 1 relates to the treatment of iconicity and systematicity. In general, I found the way the paper treated iconicity and systematicity to be sensible (following in line with, e.g. Dingemanse et al 2015, TICS), and it seems to me that the experiment with English speakers is successful at teasing these properties apart. However, considering the reviewer's thoughtful comments on this point, I believe the paper could be improved by addressing these carefully. A related issue that I found was in the definition of ‘iconicity’ in Line 64: “One possibility is that the mapping can be iconic, with aspects of form mapping onto aspects of meaning.” Without referring to ‘resemblance’ in the mapping, I don’t think this definition actually distinguishes an iconic mapping from an arbitrary one.

A second point raised by both reviewers is that the literature review may be overstating the originality of the current work in focusing on iconicity and systematicity in words. For example, Lines 211-213: “while arbitrariness, iconicity and systematicity are believed to be able to coexist, even at the level of single words, this has not been tested for iconicity and systematicity.” Please consider the suggestions on this by the reviewers. Additionally, you might consider work on universal vs. language-specific universal sound symbolism, such as reviewed by Imai & Kita (2014, Phil Transactions of the Royal Society). I think it is worthwhile to be thorough in reviewing what work there is on this topic.

Considering this all on balance, my official decision for the manuscript is ‘minor revision’.

In your revision, I highly encourage you to carefully consider all the points made by the reviewers in addition to those I have highlighted. Please also see my minor comments below. You should not feel constrained to implement all the suggestions in your paper; but if you do not, I would expect to see a clear justification for why you did not.

Minor comments:

Line 90. “Notably, much of this work has been conducted with onomatopoeic words.” My impression is that this statement is not true for several of the works cited in the previous paragraph.

Line 146. “ontology”. Is ‘ontogeny’ the word that is meant here? It seems the paper is discussing how these effects develop.

Lines 162-165. “Lyster (39) examined a 162 corpus of 9,961 French nouns, and discovered that 81% of feminine nouns, and 80% of masculine nouns, have orthographic endings that are

predictive of their grammatical gender. A predictive ending was defined as one that occurs in nouns of a certain gender at least 90% of the time.” This is confusing me. It seems that the endings are predictive 81% or 80% of the time, not 90%?

Line 475. “We observed an effect of grammatical gender only for the sharp nonwords.” Please write out what the effect was.

A final note: The subject headings, e.g. Discussion, are organized such that there are several main Discussion sections, when I think these should be subordinated to their respective Experiments.

Reviewers' comments:

Reviewer's Responses to Questions

**Comments to the Author**

1. Is the manuscript technically sound, and do the data support the conclusions?

Reviewer #1: Yes

Reviewer #2: Yes

2. Has the statistical analysis been performed appropriately and rigorously? 

Reviewer #1: Yes

Reviewer #2: Yes

3. Have the authors made all data underlying the findings in their manuscript fully available?

Reviewer #1: Yes

Reviewer #2: Yes

4. Is the manuscript presented in an intelligible fashion and written in standard English?

Reviewer #1: Yes

Reviewer #2: Yes

5. Review Comments to the Author

Reviewer #1: Review for “Is un stylo sharper than une épée? Investigating the interaction of sound symbolism and grammatical gender in English and French speakers”

General comments:

I think that this is an interesting paper which could be accepted for publication, provided especially that some more content is added to the discussion part. It is overall clearly written and well-articulated, with a sound experimental design and a good statistical approach. Data are available as well as the R code of the statistical analysis, which allows reproducibility.

Beyond minor points here and there throughout the manuscript, I have a few main concerns and suggestions:

- The authors consider iconicity and systematicity as two possibilities for non-arbitrary form-meaning mapping, which I find a bit problematic, since the first concept indeed points to the nature of the relationship between a form and a meaning, but the second one points rather to the fact that something occurs with a high frequency within a specific grammatical category. It seems to me that either non-arbitrary or arbitrary mappings could be systematic. I would rather keep the two notions on separate levels, which does not impact the core hypothesis of the paper

- More or less along the same line, the notion of linguistic category is a bit too vague to me, and this impacts the very notion of systematicity. Does linguistic means here grammatical, or it is more general?

- Both in the introduction and in the conclusion, the question is implicitly raised whether sound symbolism involves the phonological level and/or the phonetic level. This could perhaps be investigated a bit more explicitly.

- It could be recalled that the notion of gender system rests on syntactic evidence, namely agreement between the nouns and other elements of the sentence (Greville G. Corbett. 2013. Number of Genders. In: Dryer, Matthew S. & Haspelmath, Martin (eds.) The World Atlas of Language Structures Online. Leipzig: Max Planck Institute for Evolutionary Anthropology. (Available online at http://wals.info/chapter/30, Accessed on 2019-08-17.))

- As for the statistical analysis, some interactions are close to being significant predictors, and this deserves some attention, especially since statisticians have been criticizing the .05 threshold. Also, I wish to see more precisions regarding the contrasts and/or the use of ANOVA to assess the overall significance of interactions of categorical variables.

- I find the discussion(/conclusion) a bit lacking in depth. The first part is to me a bit too repetitive with respect to earlier sections (although it is good to summarize the findings and their context), and although there are some interesting points, I wish to get more “food for thought”. One suggestion I have, which I think could be analyzed statistically, would be to assess whether some subjects are much more sensitive to sensitivity than to iconicity, and vice-versa.

- A strong majority of participants to the experiments were female, which resonates with the fact that gender was at the center of the study. Although it is likely inconsequential, this could/should be discussed.

- Although the authors strongly insist on the novelty of combining “iconic” and “systematic” cues within the stimuli, previous authors have discussed the possibility that different iconic cues occurring in the same linguistic stimuli could “push in different directions”. Given the earlier comment on the nature of systematicity, and on its relationship with iconicity, one could argue that the approach is not radically new, and perhaps not original and wide-reaching enough for publication in a major scientific journal like PLOS ONE.

More detailed comments:

Line 40: “the effects of arbitrariness and non-arbitrariness in single stimuli”: it is understandable, but maybe it could be said a bit better. Maybe include a short discussion about languages with three genders like German or Serbian (not much is said about it in Serbian, although an experiment in this language is reported). Also, languages with many genders/nominal classes, e.g. a number of Bantu languages, could be mentioned.

Lines 55: the ‘need not’ is interesting, but could you provide a bit more details about why such an interpretation of Saussure’s dictum is acceptable (this dictum is very famous indeed, but finer details of Saussure’s ideas may not be familiar to all readers)

Lines 90-91…: maybe add something here about cross-cultural and cross-linguistic differences

Line 94: maybe provide an illustration or two of the concept of “existing semantic information” as soon as you mention it for the first time

Line 100: I am not convinced by the juxtaposition between iconicity and systematicity, since you may have systematic and non-systematic (non-onomatopoeic) iconicity… In other words, systematicity seems to exclude iconicity

Line 102: The concept of “linguistic category” would require more precision here. Do you mean “grammatical category”, something having to do with the structures of the language under study, or does this include semantic categories for example?

Line 108: following the previous comment, what is the distinction between concrete and abstract nouns: is it semantic? Some languages could perhaps grammatically encore the distinction between concrete and abstract nouns (I am not sure if anyone does), but there are definitely many languages which do not.

Line 112: maybe you can mention “derivational morphology” here

Line 113: once again, the juxtaposition of arbitrary, iconic or systematic is not really convincing to me.

Lune 117-118: this raises to me the question whether some subjects would be more sensitive to arbitrariness or systematicity. In turn, this raises the question whether some of your participants were more sensitive to iconicity than to systematicity, or vice-versa. This could be investigated with your data, but you did not do it. I would be interested in knowing more about that issue. It would also enrich your discussion (see my later comments about the latter)

Line 121: I would say that the articulation between the two sentences with however is not so good. Before, you are dealing with the opposition between arbitrary and systematic elements. Then you shift to iconic and systematic cues but also get rid of the memory requirement. That’s two differences, and therefore the transition is not so obvious to me

Line 128: that could be another place where to mention rich systems of genders/nominal classes, with all the accompanying intricacies.

Line 158: could you be more specific regarding the semantic differential technique?

Line 165: widespread example: maybe rephrase

Line 170: I only partially get it, could you be more explicit about the ‘portrayal in fiction’, with a possible example or two?

Line 199: words: either written or orally presented? Have these two options been contrasted?

Line 201: is it necessarily phonology, or could it be (also) phonetics? This makes sense especially with respect to some elements of your general discussion, when you insist on accents.

Line 215: stimuli rather than stimulus?

Line 221: Could the strong imbalance in favor of female participants have an impact in a task manipulating gender? Could you comment on that point somewhere? Would the comparison between males and females be possible, despite the small number of male participants?

Lines 232 to 234: could you provide here the endings you used?

Line 243: could the fact than 95% of the participants were female have had an influence on the classification? Maybe you could comment on this.

Lines 247 to 249: the same values are repeated. Could it be a mistake?

Line 266: it feels like a single practice trial was maybe not enough. Could you comment on that?

Line 270: maybe you could cite emmeans here with the other packages, rather than to cite it later

Line 272: please provide a reference for the “confirmatory approach”

Line 277: can you explain more precisely how you simplified the structure based on the suggested number of components?

Line 323: maybe you could mention some papers which have questioned the possible influence of written letters, notably Cuskley, Simner & Kirby 2015

Line 328: again, the participants were mostly female

Line 330: can you give some more details about the semantic categorization task here (you provide some later (line 368), but maybe better do it here for the sake of clarity)

Line 341 & 342: was the choice of two female voices in any way connected to the fact that the large majority of subjects were female?

Line 344: can you explain what kind of editing you did?

Line 345: I get what you did, but the connection between “one each” and the previous “two files” and “each nonword” is a bit confusing to me.

Line 382 – Table 2: the interaction between Type and Accent is nearly significant if one sticks to the infamous 5% threshold. I would suggest to pay attention to this, especially since you interpret the main effects of Type and Accent. Additionally, shouldn’t main effects be rather analyzed once non-significant interactions have been dropped from the model? And more, did you try refitting the model without the non-significant triple interaction, to check whether you would not then get a significant Type x Accent interaction?

Line 408: “that that”

Line 419: with respect to former comments, do you at least know whether the majority of participants were once again female? If the participants were more male this time, do you think one should pay attention to it?

Line 459 – Table 3: once again, the Type x Accent interaction is not far from significance – I have thus the same comments as for Table 2.

Line 530: To assess this idea of attenuation, you would have needed a third experiment with nonwords without endings suggestive of gender, then measures of effect size (OR is a measure of effect size for contingency tables or logistic regression if I am right, but the values are not that easy to interpret, compared to a measure between 0 and 1 like Cramer’s V – which we don’t readily have for complex logistic regressions with random effects) for your predictors to compare the experiments and observe a possible attenuation. Maybe that’s a perspective you could mention – in addition to the weak argument that “the results we observed speak against this notion, at least in its extreme form”.

Line 550: there have been arguments about the salience of angles and thus of sharp shapes, which seem to run counter to the hypothesis you mention. You may check De Carolis et al, 2018, in PLOS ONE, but given that I am one of the authors, this feels like I am trying to get citations for the paper – that’s not the case.

Line 579: with respect to an earlier comment, this point to the influence of phonetics in sound symbolism, and raises the question of the respective weights of phonology and phonetics in that domain. This could be an interesting elaboration.

Tables 1, 2, 3…: from what I get, you considered sum contrasts rather than treatment contrasts for your model summaries. If yes, maybe mention it explicitly. Also, I am a bit confused about the reported interactions: since you provide beta and OR, this must mean that you assess the difference between one condition (let’s say round sounding & masculine for the Type x Gender interaction, which would then mean that the base levels were sharp sounding and feminine) and the mean. This therefore does not tell you about the overall significance of the interaction, which is something a type-III ANOVA would tell you. Therefore, wouldn’t it be meaningful to (rather) report the output of Type-III ANOVAs for your models in order to better assess the significance of the interactions? (but maybe I am missing something because of the contrasts)

Reviewer #2: This very well written article uses a number of experiments to explore the interaction of systematicity (gender marking) and iconicity in determining the types of non-artbirary associations that both french and english speakers make between non-words and images varying in their curviness (the takete-maluma effect)

The article is well written, structured clearly, and interesting. It's nice that the authors don't get too bogged down in the exact same introduction that most articles about sound-symbolism/iconicity have, and instead focus on explaining the novel aspects of this work and how they relate to the aspects of language not typically discussed in this context

Overall I have almost no specific comments about the methodology or findings - they are well situated in other research and clear, and the authors do not overreach in their interpretation of those data or the claims that they make based on their findings. The authors do however state that this is the first work exploring systematicity and iconicity at the same time, which is probably not quite true. Minimally, Nielsen (2016), in his PhD thesis presents the results of an experiment where systematicity and iconicity were manipulated together. I believe Jonas Nolle from Edinburgh also has some experimental work exploring this possibility, although I am not certain if they have been published anywhere.

In the discussion, I think it would be great to hear the authors expand a bit more on whether systematicity and iconicity can be directly related to one another, rather than both being able to act as independent forces on predicting or determining word meanings.

6. PLOS authors have the option to publish the peer review history of their article (what does this mean?). If published, this will include your full peer review and any attached files.

Reviewer #1: Yes: Christophe Coupé

Reviewer #2: No

---

## [Author Response · Author response to Decision Letter 0]

28 Oct 2019

Dear Dr. Perlman:

Thank you for your decision letter of September 10th, 2019. We were very grateful for the helpful suggestions from you and the reviewers contained therein. We have carefully considered these comments and made every effort to address them in this revised version of our manuscript “Is un stylo sharper than une épée? Investigating the interaction of sound symbolism and grammatical gender in English and French speakers.”. I am resubmitting this manuscript on behalf of my co-authors, for your consideration for publication in PLOS ONE.

Below we outline the steps we have taken in this regard.

Sincerely,

David Sidhu

Editor:

Major comments:

The main point of concern of Reviewer 1 relates to the treatment of iconicity and systematicity. In general, I found the way the paper treated iconicity and systematicity to be sensible (following in line with, e.g. Dingemanse et al 2015, TICS), and it seems to me that the experiment with English speakers is successful at teasing these properties apart. However, considering the reviewer's thoughtful comments on this point, I believe the paper could be improved by addressing these carefully. 

To try and address any possible misunderstanding, we now explicitly state that we are working within the paradigm set out in Dingemanse et al. (2015). We also note in the General Discussion that our methods were designed to isolate iconicity and systematicity.

A related issue that I found was in the definition of ‘iconicity’ in Line 64: “One possibility is that the mapping can be iconic, with aspects of form mapping onto aspects of meaning.” Without referring to ‘resemblance’ in the mapping, I don’t think this definition actually distinguishes an iconic mapping from an arbitrary one.

Thank you for pointing this out, we have now added “via resemblance” to the definition.

A second point raised by both reviewers is that the literature review may be overstating the originality of the current work in focusing on iconicity and systematicity in words. For example, Lines 211-213: “while arbitrariness, iconicity and systematicity are believed to be able to coexist, even at the level of single words, this has not been tested for iconicity and systematicity.” Please consider the suggestions on this by the reviewers. Additionally, you might consider work on universal vs. language-specific universal sound symbolism, such as reviewed by Imai & Kita (2014, Phil Transactions of the Royal Society). I think it is worthwhile to be thorough in reviewing what work there is on this topic.

Thank you for pointing this out. We now have a paragraph in the General Discussion talking about how these could interact, including Imai and Kita’s theory that some sound symbolic associations are only apparent to individuals if they are attested to in the iconic words of that speaker’s language. This is presented as an example of iconicity and systematicity coexisting. In addition, we have rephrased the sentence you mention, to read: “…this has yet to be fully examined”, to make a less strong claim about the originality of this work. 

Minor comments:

Line 90. “Notably, much of this work has been conducted with onomatopoeic words.” My impression is that this statement is not true for several of the works cited in the previous paragraph.

Thank you for catching this. We have removed the line in question and have rephrased the section that follows slightly: “Despite this work, the extent to which non-onomatopoeic iconicity can affect processing is still somewhat unclear. Studies that have directly examined sound symbolic associations such as the maluma/takete effect in existing language have been equivocal”

Line 146. “ontology”. Is ‘ontogeny’ the word that is meant here? It seems the paper is discussing how these effects develop.

Thank you, have switched to the simpler “development”.

Lines 162-165. “Lyster (39) examined a corpus of 9,961 French nouns, and discovered that 81% of feminine nouns, and 80% of masculine nouns, have orthographic endings that are

predictive of their grammatical gender. A predictive ending was defined as one that occurs in nouns of a certain gender at least 90% of the time.” This is confusing me. It seems that the endings are predictive 81% or 80% of the time, not 90%?

We have removed the definition of what constitutes a predictive ending (i.e., an ending is predictive if 90% of the words in which it occurs are feminine (or masculine)), which should clear this up. Just for interest’s sake, to clear up the confusion: 

An ending is predictive if 90% of the words in which it occurs are feminine (or masculine). 80% of feminine words have such endings. 

Line 475. “We observed an effect of grammatical gender only for the sharp nonwords.” Please write out what the effect was.

We have done so.

A final note: The subject headings, e.g. Discussion, are organized such that there are several main Discussion sections, when I think these should be subordinated to their respective Experiments.

Have fixed this, thanks!

Reviewer #1: 

General comments:

- The authors consider iconicity and systematicity as two possibilities for non-arbitrary form-meaning mapping, which I find a bit problematic, since the first concept indeed points to the nature of the relationship between a form and a meaning, but the second one points rather to the fact that something occurs with a high frequency within a specific grammatical category. It seems to me that either non-arbitrary or arbitrary mappings could be systematic. I would rather keep the two notions on separate levels, which does not impact the core hypothesis of the paper

We are using the framework suggested in Dingemanse et al. 2015, which is quite common (206 citations), though certainly there may be other ways to construe this. We have now made it clear that we are using this framework by saying “using the framework suggested by…”. Under this framework, systematicity is a form of non-arbitrariness, and so a systematic mapping could not be arbitrary. Systematic mappings could, however, be iconic, as these categories are not mutually exclusive. We have added discussion of this to the General Discussion.

- More or less along the same line, the notion of linguistic category is a bit too vague to me, and this impacts the very notion of systematicity. Does linguistic means here grammatical, or it is more general?

The majority of examples studied include syntactic categories (e.g., nouns vs verbs; open vs. closed class verbs) though Dingemanse et al. do note that there could even be form patterns correlating with semantic factors such as concreteness. Thus we now say “words belonging to the same syntactic (or even semantic) category”. 

- Both in the introduction and in the conclusion, the question is implicitly raised whether sound symbolism involves the phonological level and/or the phonetic level. This could perhaps be investigated a bit more explicitly.

We have retained the term “phonology” in the intro, as we intend to use it as it seems to be used in psychology (i.e., referring to the auditory properties of a word, rather than its orthography or meaning). We have now added a paragraph to the General Discussion talking about the complementary effects of acoustic features and phonemic categories. Here we mention that these two concepts would be termed “phonetics” and “phonology” in linguistics.

- It could be recalled that the notion of gender system rests on syntactic evidence, namely agreement between the nouns and other elements of the sentence (Greville G. Corbett. 2013. Number of Genders. In: Dryer, Matthew S. & Haspelmath, Martin (eds.) The World Atlas of Language Structures Online. Leipzig: Max Planck Institute for Evolutionary Anthropology. (Available online at http://wals.info/chapter/30, Accessed on 2019-08-17.))

Have added this, thanks.

- As for the statistical analysis, some interactions are close to being significant predictors, and this deserves some attention, especially since statisticians have been criticizing the .05 threshold. Also, I wish to see more precisions regarding the contrasts and/or the use of ANOVA to assess the overall significance of interactions of categorical variables.

Thank you this suggestion. We are in full agreement that effects near the .05 cutoff should not be entirely disregarded. We now offer some interpretation of these effects in the General Discussion (i.e., increasing the roundness of sharp-sounding nonwords with a French accent may have attenuated effects of nonword type in that condition; thus a stronger effect of nonword type for English nonwords). 

Additionally, we have looked into the question of whether an ANOVA is necessary to follow up on the interaction term in a glmer() model. We were directed to this paper which suggests it is not necessary. https://arxiv.org/abs/1807.10451

- I find the discussion(/conclusion) a bit lacking in depth. The first part is to me a bit too repetitive with respect to earlier sections (although it is good to summarize the findings and their context), and although there are some interesting points, I wish to get more “food for thought”. One suggestion I have, which I think could be analyzed statistically, would be to assess whether some subjects are much more sensitive to sensitivity than to iconicity, and vice-versa.

As mentioned, we have added some content on the effects of phonology vs. phonetics, as well as the interplay of iconicity and systematicity. We also discuss how these effects compare to similar experiments in the field when nonwords do not have grammatical endings.

Regarding the point of comparing iconicity and systematicity sensitivity, thank you for this very interesting suggestion! We think the ideal way to do this would be to compare each subject’s random slope for effects of gender and shape. These should be interpreted with caution, as the analytical approach we used suggests that inclusion of both random effects was not warranted. For this reason we do not add these results to the paper, though we include them here for the sake of interest.

Below you will find each subject’s random slope for shape and gender, for each of the experiments we ran. As you can see, nearly every participant showed a stronger effect of shape than gender. Though there were a few cases of the opposite, this seemed to be because of a smaller shape effect in a particular participant. Interestingly, the effect of gender ending seemed to behave in the opposite direction in the experiment with English speakers, supporting the position that they did not derive gender from these endings. It seems that on the whole, iconicity was a stronger cue than systematicity for nearly all participants.

(Graphs in uploaded cover letter)

- A strong majority of participants to the experiments were female, which resonates with the fact that gender was at the center of the study. Although it is likely inconsequential, this could/should be discussed.

In Sidhu & Pexman 2015 Experiment 1a, we had participants associate male and female names with round or sharp-sounding phonemes, with round or sharp silhouettes. We found an effect of both name sound and gender. Notably, in that experiment, we had a roughly even split of participant gender (23 males; 30 females) and so examined whether participant gender affected either effect. It did not. Thus there is some evidence of participant gender not affecting pairings between gendered targets and round/sharp shapes.

Unfortunately we did not have enough males in the current experiments to check whether the same would hold true here as well. Nevertheless, we have added a line in the General Discussion mentioning that future research could examine individual differences playing a role in grammatical gender associations.

- Although the authors strongly insist on the novelty of combining “iconic” and “systematic” cues within the stimuli, previous authors have discussed the possibility that different iconic cues occurring in the same linguistic stimuli could “push in different directions”. Given the earlier comment on the nature of systematicity, and on its relationship with iconicity, one could argue that the approach is not radically new, and perhaps not original and wide-reaching enough for publication in a major scientific journal like PLOS ONE.

We don’t intend to claim that our approach is radically new. However, after the clarification that we are working within a specific paradigm of arbitrariness/iconicity/systematicity, we do believe that examining the iconicity and systematicity within single stimuli is novel (at least in terms of how these concepts are defined in this paradigm). Nevertheless, we have tempered claims throughout the manuscript with regards to the novelty of this work.

More detailed comments:

Line 40: “the effects of arbitrariness and non-arbitrariness in single stimuli”: it is understandable, but maybe it could be said a bit better. Maybe include a short discussion about languages with three genders like German or Serbian (not much is said about it in Serbian, although an experiment in this language is reported). Also, languages with many genders/nominal classes, e.g. a number of Bantu languages, could be mentioned.

Thanks, we have tried to clarify with “…effects of arbitrary and non-arbitrary mappings contained in a single stimulus…”

We do mention evidence that this seems to be stronger in languages with two than three genders. We aren’t inclined to add descriptions of grammatical genders in other languages as it might disrupt the flow of the article.

Lines 55: the ‘need not’ is interesting, but could you provide a bit more details about why such an interpretation of Saussure’s dictum is acceptable (this dictum is very famous indeed, but finer details of Saussure’s ideas may not be familiar to all readers)

Rather than going into this, we have removed reference to Saussure’s arbitrariness, in part because we are sympathetic to the view expressed in Hutton (1989) that Saussure was talking about the relationship between mentalistic concepts rather than the phonological form of a word and its referent in the world. We now use Hockett’s (1963) notion of arbitrariness, which is much more straight forward.

Lines 90-91…: maybe add something here about cross-cultural and cross-linguistic differences

We have not included this here as it did not seem sufficiently relevant to the main focus. However in the General Discussion we mention differential effects of phonetics and phonology to cross-linguistic differences.

Line 94: maybe provide an illustration or two of the concept of “existing semantic information” as soon as you mention it for the first time

We have added a line to clarify we mean that when a word has meaning, retrieving this meaning could interfere with the activation of sound symbolic associations.

Line 100: I am not convinced by the juxtaposition between iconicity and systematicity, since you may have systematic and non-systematic (non-onomatopoeic) iconicity… In other words, systematicity seems to exclude iconicity

We fully agree that these are not mutually exclusive, and we now make that clear in our definition of each. In addition, we give two examples in which they might interact. Even in these cases, however, there are two independent forces at work: iconicity and systematicity. We believe it makes sense to treat them as separate forces that can combine. Here, however, we are examining them separately, because we isolate each effect (i.e., cues to grammatical gender are not iconic).

Line 102: The concept of “linguistic category” would require more precision here. Do you mean “grammatical category”, something having to do with the structures of the language under study, or does this include semantic categories for example?

We have clarified that we mean syntactic and semantic categories. 

Line 108: following the previous comment, what is the distinction between concrete and abstract nouns: is it semantic? Some languages could perhaps grammatically encode the distinction between concrete and abstract nouns (I am not sure if anyone does), but there are definitely many languages which do not.

Yes this was based on concreteness and imageability ratings. We haven’t added this detail to the paper but would be happy to do so.

Line 112: maybe you can mention “derivational morphology” here

Added mention of this, thanks.

Line 113: once again, the juxtaposition of arbitrary, iconic or systematic is not really convincing to me.

See earlier comments.

Lune 117-118: this raises to me the question whether some subjects would be more sensitive to arbitrariness or systematicity. In turn, this raises the question whether some of your participants were more sensitive to iconicity than to systematicity, or vice-versa. This could be investigated with your data, but you did not do it. I would be interested in knowing more about that issue. It would also enrich your discussion (see my later comments about the latter)

See earlier comment.

Line 121: I would say that the articulation between the two sentences with however is not so good. Before, you are dealing with the opposition between arbitrary and systematic elements. Then you shift to iconic and systematic cues but also get rid of the memory requirement. That’s two differences, and therefore the transition is not so obvious to me

Have removed mention of the memory aspect.

Line 128: that could be another place where to mention rich systems of genders/nominal classes, with all the accompanying intricacies.

We attempt to address this to some extent in the section mentioning that effects of grammatical gender can vary based on specific features (e.g., the number of grammatical categories in a language). 

Line 158: could you be more specific regarding the semantic differential technique?

Have added a description of this method.

Line 165: widespread example: maybe rephrase

We have changed this to “prevalent”.

Line 170: I only partially get it, could you be more explicit about the ‘portrayal in fiction’, with a possible example or two?

We have now added the specific example given by Boroditsky et al., about objects being personified differently in fairy tales or poetry based on the grammatical gender.

Line 199: words: either written or orally presented? Have these two options been contrasted?

We are aware of this in studies of sound symbolism, however not in the context of grammatical gender.

Line 201: is it necessarily phonology, or could it be (also) phonetics? This makes sense especially with respect to some elements of your general discussion, when you insist on accents.

See earlier comments.

Line 215: stimuli rather than stimulus?

We think stimuli is correct.

Line 221: Could the strong imbalance in favor of female participants have an impact in a task manipulating gender? Could you comment on that point somewhere? Would the comparison between males and females be possible, despite the small number of male participants?

See earlier comment.

Lines 232 to 234: could you provide here the endings you used?

Thank you for this suggestion, we have added the table.

Line 243: could the fact than 95% of the participants were female have had an influence on the classification? Maybe you could comment on this.

See earlier comment.

Lines 247 to 249: the same values are repeated. Could it be a mistake?

Thank you for flagging this, however it is not a mistake.

Line 266: it feels like a single practice trial was maybe not enough. Could you comment on that?

Because this is such a simple task, we think that a single practice trial is enough. The experimenter was also present during the trial in case the participant had any questions.

Line 270: maybe you could cite emmeans here with the other packages, rather than to cite it later

We considered this, but would rather keep it later so it’s clear what calculation that function is performing.

Line 272: please provide a reference for the “confirmatory approach”

We have added reference to a chapter discussing this.

Line 277: can you explain more precisely how you simplified the structure based on the suggested number of components?

We now explain that this was accomplished by removing the random slope for the highest order effect with the lowest amount of associated variance.

Line 323: maybe you could mention some papers which have questioned the possible influence of written letters, notably Cuskley, Simner & Kirby 2015

Thank you for suggestion, we have added this.

Line 328: again, the participants were mostly female

See previous comment.

Line 330: can you give some more details about the semantic categorization task here (you provide some later (line 368), but maybe better do it here for the sake of clarity)

Added mention that the task was about “does this word refer to a living or a non-living thing?”.

Line 341 & 342: was the choice of two female voices in any way connected to the fact that the large majority of subjects were female?

We selected voices based on those that were the most intelligible. 

Line 344: can you explain what kind of editing you did?

Audio recordings were created with Sound Tap Streaming Audio Recorder version 2.31, and sound files were edited with WavePad Sound Editor version 5.96. Editing simply consisted of removing the silence before and after the nonword.

Line 345: I get what you did, but the connection between “one each” and the previous “two files” and “each nonword” is a bit confusing to me.

Thank you for flagging this, we have changed it to: “versions of each nonword: one with a French and one with an English pronunciation” which should clear it up.

Line 382 – Table 2: the interaction between Type and Accent is nearly significant if one sticks to the infamous 5% threshold. I would suggest to pay attention to this, especially since you interpret the main effects of Type and Accent. Additionally, shouldn’t main effects be rather analyzed once non-significant interactions have been dropped from the model? And more, did you try refitting the model without the non-significant triple interaction, to check whether you would not then get a significant Type x Accent interaction?

See earlier comment re: ANOVA.

Thank you for the suggestion of rerunning the model without the threeway interaction. The p value for the interaction is p = .0523 (after developing random effects in the way we describe elsewhere). Nevertheless, we agree it’s worth interpreting and now mention it in the section about accent, suggesting that the French accent may have made sharp words seem rounder and thus attenuated the effect. We continue with the model including all interactions. Since we have used effects coding, lower order effects can be interpreted in the presence of higher order interactions (as these effects will be at the average of all other terms).

Line 408: “that that”

Corrected.

Line 419: with respect to former comments, do you at least know whether the majority of participants were once again female? If the participants were more male this time, do you think one should pay attention to it?

While that would have been interesting, the researcher who ran this study believes they were again mostly female.

Line 459 – Table 3: once again, the Type x Accent interaction is not far from significance – I have thus the same comments as for Table 2.

This effect without the threeway interaction is now p = .071. See earlier comment about our interpretation.

Line 530: To assess this idea of attenuation, you would have needed a third experiment with nonwords without endings suggestive of gender, then measures of effect size (OR is a measure of effect size for contingency tables or logistic regression if I am right, but the values are not that easy to interpret, compared to a measure between 0 and 1 like Cramer’s V – which we don’t readily have for complex logistic regressions with random effects) for your predictors to compare the experiments and observe a possible attenuation. Maybe that’s a perspective you could mention – in addition to the weak argument that “the results we observed speak against this notion, at least in its extreme form”.

Thank you for this point. We now provide some anecdotal comparisons of the coefficients observed here to those of similar previous studies in which stimuli do or do not have other associated information. We also mention that while sound symbolism effects will emerge when stimuli have other existing information associated with them, there may be some attenuation. We mention that future research directly comparing these kinds of stimuli would be needed, however.

Line 550: there have been arguments about the salience of angles and thus of sharp shapes, which seem to run counter to the hypothesis you mention. You may check De Carolis et al, 2018, in PLOS ONE, but given that I am one of the authors, this feels like I am trying to get citations for the paper – that’s not the case.

Have added reference, thanks.

Line 579: with respect to an earlier comment, this point to the influence of phonetics in sound symbolism, and raises the question of the respective weights of phonology and phonetics in that domain. This could be an interesting elaboration.

See earlier comments.

Tables 1, 2, 3…: from what I get, you considered sum contrasts rather than treatment contrasts for your model summaries. If yes, maybe mention it explicitly. Also, I am a bit confused about the reported interactions: since you provide beta and OR, this must mean that you assess the difference between one condition (let’s say round sounding & masculine for the Type x Gender interaction, which would then mean that the base levels were sharp sounding and feminine) and the mean. This therefore does not tell you about the overall significance of the interaction, which is something a type-III ANOVA would tell you. Therefore, wouldn’t it be meaningful to (rather) report the output of Type-III ANOVAs for your models in order to better assess the significance of the interactions? (but maybe I am missing something because of the contrasts)

See earlier comment.

Reviewer #2: 

The authors do however state that this is the first work exploring systematicity and iconicity at the same time, which is probably not quite true. Minimally, Nielsen (2016), in his PhD thesis presents the results of an experiment where systematicity and iconicity were manipulated together. I believe Jonas Nolle from Edinburgh also has some experimental work exploring this possibility, although I am not certain if they have been published anywhere.

We now cite Nielsen 2016 in intro, with the distinction being made that in this work the iconic and systematic cues are separate, unlike in Nielsen 2016. We have also tempered our claims about the originality of this work.

In the discussion, I think it would be great to hear the authors expand a bit more on whether systematicity and iconicity can be directly related to one another, rather than both being able to act as independent forces on predicting or determining word meanings.

Thank you for this suggestion, we have now added some discussion of this interplay.

---

## [Editor Report · Decision Letter 1]

11 Nov 2019

Is un stylo sharper than une épée? Investigating the interaction of sound symbolism and grammatical gender in English and French speakers

PONE-D-19-19956R1

Dear Dr. Sidhu,

We are pleased to inform you that your manuscript has been judged scientifically suitable for publication and will be formally accepted for publication once it complies with all outstanding technical requirements.

With kind regards,

Marcus Perlman, Ph.D

Academic Editor

PLOS ONE

Additional Editor Comments (optional):

Thank you for your detailed revisions and thoughtful responses to the comments of the reviwers (and me). I am pleased to accept the paper for publication at PLOS ONE -- congratulations.
---

## [Editor Report · Acceptance letter]

20 Nov 2019

PONE-D-19-19956R1 

Is un stylo sharper than une épée? Investigating the interaction of sound symbolism and grammatical gender in English and French speakers 

Dear Dr. Sidhu:

I am pleased to inform you that your manuscript has been deemed suitable for publication in PLOS ONE. Congratulations! Your manuscript is now with our production department. 

With kind regards,

on behalf of

Dr. Marcus Perlman 

Academic Editor

PLOS ONE